# Splinting in horizontal root fractures: A Bayesian network meta-analysis

Tatiana Carvalho Kowaltschuk[1,2,☯,*,¤a], Flávio Magno Gonçalves[2,☯,¤b],
Bianca Marques de Mattos de Araujo[1,2,☯,¤b], Natália Teixeira da Silva Cruz[2,‡,¤b],
Angela Graciela Deliga Schroder[3,‡], Bruna Belz Antoniazzi[4,‡,¤b], Everdan Carneiro[1,¤a],
Marisa Nogueira Alencar[1,‡,¤a], Ulisses Xavier da Silva-Neto[1,‡,¤a], Odilon Guariza-Filho[2,5,☯,¤b],
Cristiano Miranda de Araujo[2,3,☯,¤b], Vânia Portela Ditzel Westphalen[1,☯,¤a]

1 Postgraduate Program in Dentistry, Department of Endodontics, Pontifícia Universidade Católica do
Paraná, Curitiba, Paraná, Brazil, 2 Center for Advanced Studies in Systematic Review and Meta-Analysis
- NARSM, Curitiba, Paraná, Brazil, 3 Postgraduate Program in Dentistry, Department of Orthodontics,
Tuiuti University of Paraná, Curitiba, Paraná, Brazil, 4 Undergraduate Program in Dentistry, Department
of Endodontics, Pontifícia Universidade Católica do Paraná, Curitiba, Paraná, Brazil, 5 Postgraduate
Program in Dentistry, Department of Orthodontics, Pontifícia Universidade Católica do Paraná, Curitiba,
Paraná, Brazil

☯ These authors contributed equally to this work.
‡ NTDC, AGDS, BBA, EC, MNA and UXDN also contributed equally to this work.
¤a Current Address: Postgraduate Program in Dentistry, Department of Endodontics, Pontifícia
Universidade Católica do Paraná, Curitiba, Paraná, Brazil
¤b Current Address: Center for Advanced Studies in Systematic Review and Meta-Analysis – NARSM,
Curitiba, Paraná, Brazil
* tatiana.kowa@hotmail.com

## Abstract

### Objective

This study aims to evaluate which splinting method offers the most favorable prognosis in cases of dental trauma involving horizontal intra-alveolar root fractures.

### Materials and methods

Search strategies were adapted for seven electronic databases and gray literature to identify studies evaluating the prognosis of horizontal intra-alveolar root fractures based on the type of splint used. The risk of bias in the included studies was assessed using the Joanna Briggs Institute Critical Appraisal tools. A Bayesian network meta-analysis was conducted.

### Results

A total of 3,174 references were retrieved, of which six studies met the inclusion criteria. No significant differences were found between the types of splints used and the healing outcomes of horizontally fractured intra-alveolar roots, including comparisons to cases without fixation. Across all studies, the degree of fragment displacement was a key factor influencing healing: the less displacement, the better the prognosis.

**Data availability statement:** All relevant data are within the manuscript and its Supporting Information files.

**Funding:** The author(s) received no specific funding for this work.

**Competing interests:** The authors have declared that no competing interests exist.

Other treatment-related variables analyzed by the authors also played a role in determining tooth prognosis.

## Conclusions

There may be no difference regarding the type of splint used when considering the healing of horizontal intra-alveolar root fractures; however, other factors may have influenced this outcome.

## Clinical Relevance

Understanding the fracture characteristics and the type of splinting intervention employed is essential for ensuring appropriate treatment in such cases.

## Introduction

Dental trauma can significantly impact the quality of life of children and adolescents. Most traumatic dental injuries involve anterior teeth, which can lead to limitations such as difficulty biting and speaking clearly, as well as aesthetic and social problems when showing their teeth. Clinical investigations indicate that 16% to 30% of children and adolescents have experienced some type of dental trauma [1], among which 0.5–7% were intra-alveolar root fractures in the permanent dentition [2–5]. Horizontal intra-alveolar root fracture is a trauma that affects the cementum, dentin, and pulp [3], and is classified as apical third, middle third, and cervical third fracture [3–6].

The diagnosis of horizontal root fracture is based on clinical findings, including tooth mobility, a positive percussion test, bleeding from the gingival sulcus, inconclusive results from a pulp sensibility test, and crown discoloration. These findings should be correlated with radiographic examinations [3,5]. Periapical and occlusal radiographs are the standard imaging modalities recommended by the IADT (International Association of Dental Traumatology). However, intraoral radiographs have limitations due to their two-dimensional nature and the superimposition of anatomical structures [7]. New technologies, such as cone-beam computed tomography (CBCT) specifically designed for dental applications, can overcome these limitations [8,9]. In cases of complex intra-alveolar root fractures, CBCT imaging can be essential, providing highly detailed views of the trauma and offering additional information for accurate treatment planning.

The healing process begins at the moment of trauma and depends on factors such as the patient's age, tooth mobility, root development, fracture location, fragment separation, and the time between the trauma and treatment [10–12]. Another critical factor is the fracture location; the more apical the fracture, the better the prognosis [3,4]. The healing of horizontal intra-alveolar root fracture depends on the integrity of the dental pulp and the fracture level. If the pulp is intact and adequately vascularized, odontoblasts can form dentin, uniting the fragments and forming a dental callus. Cementum is formed by cells derived from the periodontal ligament. This process is referred to as healing by hard tissue repair. If the fragments are separated,

the vascular supply is compromised, there is healthy pulp tissue, and no contamination is present, connective tissue is formed, eventually leading to the formation of cement. In this case, the fragments remain stable even if they stay separated. This healing process involves the interposition of connective tissue alone or connective tissue along with bone. When tissue contamination occurs and leads to pulp necrosis, an inflammatory reaction begins, stimulating the production of granulation tissue between the two fragments. This response is referred to as 'non-healing' or 'non-repair' [3,13].

Splinting is part of the standard treatment recommended by the IADT for cases of horizontal intra-alveolar root fractures. Various types of splints are available, and over the years, new designs have been developed to treat this type of dental trauma. The main difference among splints lies in their flexibility; some allow more movement of the tooth and surrounding soft tissues, while others are rigid and restrict movement [14].

In a previous review, Kahler and Heithersay (2008) observed that the type of trauma (dislocation, avulsion or root fracture) had a greater influence on treatment outcomes than immobilization factors (splint type and duration). However, different types of splint were associated with varying frequencies of pulp necrosis and pulp canal obliteration (pulp calcification) [15]. No systematic reviews identified in the scientific literature that focus specifically on horizontal intra-alveolar root fractures and quantify the differences between splint types. Thus, there is a need for a systematic review to assess the prognosis of these fractures according to the splinting method used.

Therefore, this study aimed to evaluate which splinting method is associated with better prognosis in cases of dental trauma involving horizontal intra-alveolar root fractures.

## Materials and methods

To conduct this systematic review, the PRISMA-NMA (PRISMA Extension Statement for Reporting of Systematic Reviews Incorporating Network Meta-analyses of Health Care Interventions) guidelines were followed. The methods and results of systematic reviews should be reported in sufficient detail to allow users to assess the trustworthiness and applicability of the review findings. The Preferred Reporting Items for Systematic reviews and Meta-Analyses (PRISMA) statement was developed to facilitate transparent and complete reporting of systematic reviews and has been updated (PRISMA 2020) to reflect recent advances in systematic review methodology and terminology [16].

### Protocol and registration

The protocol was registered on the PROSPERO website (International Prospective Register of Systematic Reviews – University of York – Centre for Reviews and Dissemination - CRD42021244955).

### Eligibility criteria

The PICOS acronym was used to establish the eligibility criteria for the following research question: "Which type of splint has a better prognosis in cases of horizontal intra-alveolar root fracture?":

**P (participants):** patients who experienced dental trauma involving horizontal intra-alveolar root fractures in permanent teeth;

**I (intervention):** teeth that were treated with any type of splinting;

**C (comparison):** different types of splinting;

**O (outcomes):**

- Primary outcome: healing;

- Secondary outcomes: type of healing (hard tissue formation, connective tissue formation, granulation tissue formation, or pulp necrosis);

**S (study design):** cross-sectional observational studies, cohort, case-control, randomized, and pseudo-randomized or quasi-randomized clinical trials.

Based on these criteria, the following inclusion and exclusion criteria were defined:

a) **Population:** Studies were included if the sample consisted of patients who had suffered dental trauma resulting in horizontal intra-alveolar root fractures in permanent teeth, with either complete or incomplete root formation, regardless of patient age. Studies were excluded if the trauma did not involve a horizontal intra-alveolar root fracture or if the fracture occurred in a primary tooth. There were no exclusions based on gender or ethnicity.

b) **Intervention/exposition:** Studies were included if the affected teeth were treated using rigid, semi-rigid, or flexible splints. Studies were excluded if the teeth were orthodontically extracted or if splinting was combined with another type of intervention unrelated to the splinting method.

c) **Comparison:** Only studies that compared at least two types of splints were included. Studies without any type of comparison were excluded.

d) **Outcomes:** Studies were included if they reported data on the healing of horizontally fractured teeth, tooth survival (permanence in the oral cavity), or the need for endodontic treatment (pulpal status). Studies that did not evaluate these outcomes or that presented incomplete data were excluded.

e) **Study design:** Included studies comprised cross-sectional observational studies, cohort studies, case-control studies, randomized clinical trials, and pseudo- or quasi-randomized clinical trials. Reviews, case reports, case series, letters to the editor, expert opinions, and books were excluded. No exclusions were made based on language or year of publication.

## Information sources and search strategy

The search strategy involved identifying and refining appropriate word combinations and truncations for the following seven databases: Pubmed/Medline, Scopus, Web of Science, LIVIVO, Latin American and Caribbean Literature on Health Sciences (LILACS), Embase, and the Cochrane Library. In addition, gray literature search was also performed through Google Scholar, Proquest, and Open Grey (S1 Appendix).

References retrieved from the searches were also manually screened, and an expert was consulted to recommend relevant studies with potential for inclusion. EndNote® software (Clarivate, Philadelphia, United States of America) was used to manage references and remove duplicates. All searches were performed on April 14, 2021 and were updated on April 02, 2025.

## Study selection

The studies were selected in two stages. In the first stage, two reviewers independently screened the titles and abstracts of the retrieved papers (Phase 1). Studies that did not meet the inclusion criteria were excluded. In the second stage (Phase 2), the same reviewers independently read the full texts of the selected papers. Disagreements were discussed during a consensus meeting. If disagreement persisted between the two reviewers, a third reviewer was consulted to make the final decision. When the available data were insufficient to determine a study's eligibility, up to three attempts (at one-week intervals) were made to contact the corresponding author via email.

The Rayyan website (http://rayyan.qcri.org) was used to ensure independent screening and reviewer blinding during both phases of the selection process. A team member that was not involved in the selection process acted as moderator. Prior to the screening phase, reviewer calibration was performed using a partial literature search, through which the Kappa Coefficient of Agreement was calculated. Screening in Phases 1 and 2 only began after achieving a Kappa value > 0.7, indicating good agreement between reviewers.



### Data collection process

The following data were collected: study characteristics (authors, year of publication, and country), population characteristics (sample size, sex, and age), assessment characteristics (splint type), outcome characteristics (main results), and main conclusions. Two reviewers independently extracted data from the included studies.

### Data items

The frequency of events related to fracture healing, type of healing, pulp necrosis, and tooth survival were extracted for each splint type evaluated. Potential confounding factors, such as the mean age and specific third in which the horizontal intra-alveolar root fracture occurred fracture occurred, were also recorded.

### Network geometry

Each node in the network represented a splint type used for the treatment of horizontal intra-alveolar root fractures. The connecting lines indicated available direct comparisons. Data from all interventions directly related to splinting were tabulated.

### Risk of bias within individual studies

The Joanna Briggs Institute Critical Appraisal tool [17] was used to assess the risk of bias in the included studies. Studies were classified as having a high risk of bias (< 49% "Yes" responses), moderate risk (50–69% "Yes"), or low risk (> 70% "Yes") [18]. Two reviewers independently assessed the risk of bias. In case of disagreement that could not be resolved through discussion, a third reviewer was consulted for the final decision.

### Summary measures and planned methods of analysis

The primary outcome assessed was intra-alveolar root fracture healing, according to the criteria established by Andreasen and Hjørting-Hansen (1967). Healing types 1, 2, and 3 were considered successful healing, while type 4 was considered as failure [19]. Data were extracted for each treatment type. Studies that compared more than two splint types were included in the quantitative synthesis separately, providing more than one effect size. Since the extracted measures were binary data, the relative risk (RR) for the outcome of interest was calculated using traditional meta-analysis by including multiple different pairwise comparisons with respective 95% confidence intervals (95% CI). When studies shared the same group, only one was retained to avoid duplication of data and effect sizes within the same analysis. To satisfy the transitivity assumption, all included studies evaluated the use of splinting in cases of horizontal intra-alveolar root fracture. Splint types were grouped according to their reported characteristics, ensuring consistent modifier variables across different comparisons.

### Network synthesis

A mixed treatment comparison (MTC) was conducted, as the evidence network included more than two interventions, with some comparisons made directly and others indirectly. The network meta-analysis was performed using Bayesian statistical methods, incorporating both random and fixed effects, with a binomial likelihood and a logarithmic link function. Four Markov chains were run for each model (n.chain = 4) to improve convergence and statistical stability, and to obtain the posterior distribution of the logarithm of the risk ratio. A uniform distribution (dunif) was used for the between-sudy scale parameter. Fifty thousand iterations with a thinning interval of 30 were employed after 5000 burn-in iterations per chain. The inverse-variance weighted method was applied, with variance estimated using the DerSimonian and Laird method.

 To ensure the reliability of the analysis, diagnostic assessments were performed using autocorrelation plots, traceplots, and Gelman-Rubin statistics to assess model convergence and consistency. The Deviance Information Criterion (DIC)

was used to compare the random and fixed effects models, with preference given to the model with the lower DIC value. Based on this criterion, the random effects model was selected. Heterogeneity was assessed using the Higgins inconsistency index ($I^2$). The results of the network meta-analysis were reported with their respective credibility intervals (CrI95%). All statistical analyses and graphical representations were performed using R statistical software, version 4.0.2 (R Foundation for Statistical Computing).

### Assessment of inconsistency

Consistency between direct and indirect comparisons within the Bayesian network was assessed using node-splitting analysis, which evaluated the robustness of comparisons involving pairs of treatments supported by both types of evidence [20].

## Results

### Study selection

In Phase 1 of the study selection process, 3,174 papers were retrieved from the seven selected databases. After removing duplicates, 1,830 papers remained. A flowchart detailing the identification, inclusion, and exclusion process is presented in Fig 1. Following the completion of Phase 1, 26 papers were selected for full-text reading (Phase 2). Of these, 21 were excluded leaving five studies for inclusion in the qualitative synthesis (S2 Appendix). After manually screening the reference lists, one additional study was included.

### Study characteristics

According to their design, all included studies were cross-sectional observational. Each study compared different splint types with varying levels of flexibility. The evaluation involved radiographic analysis and classification according to the type of healing. Although no language restrictions were applied during study selection, only studies published in English were identified in the literature. The main characteristics of the included papers are shown in Table 1.

### Summary of network geometry and structure

The evidence network consisted of eight different splint types and one group without any form of treatment (no fixation). Each of these groups was represented as a node in the network (Fig 2), with node size corresponding to the number of studies using each intervention. The lines connecting the nodes represented direct comparisons between the different splint types in terms of healing outcomes. The number of comparisons was reflected in the thickness of each line.

Out of the six studies included in the analysis, the groups with the most direct pairwise comparisons were those involving cap splints, orthodontic bands, arch bars, and the 'no splint' group.

### Risk of bias in studies

All studies were classified as having a low risk of bias. However, three studies did not control confounding factors [19,21,22]. All studies showed uncertainty regarding the method of outcome assessment (Fig 3). In addition, all studies used periapical radiographs to perform the evaluation, which may introduce distortion in the results, as these provide a two-dimensional representation of a three-dimensional structure.

### Results of individual studies

Several variables may influence the outcome. Among them are the degree of fragment displacement, whether immediate splinting was performed, the degree of fracture reduction, apical foramen diameter, periodontal health, presence of restorations, use or non-use of antibiotics, fragment mobility, stage of root development, tooth sensitivity after trauma, and

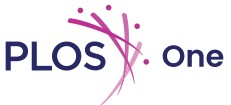

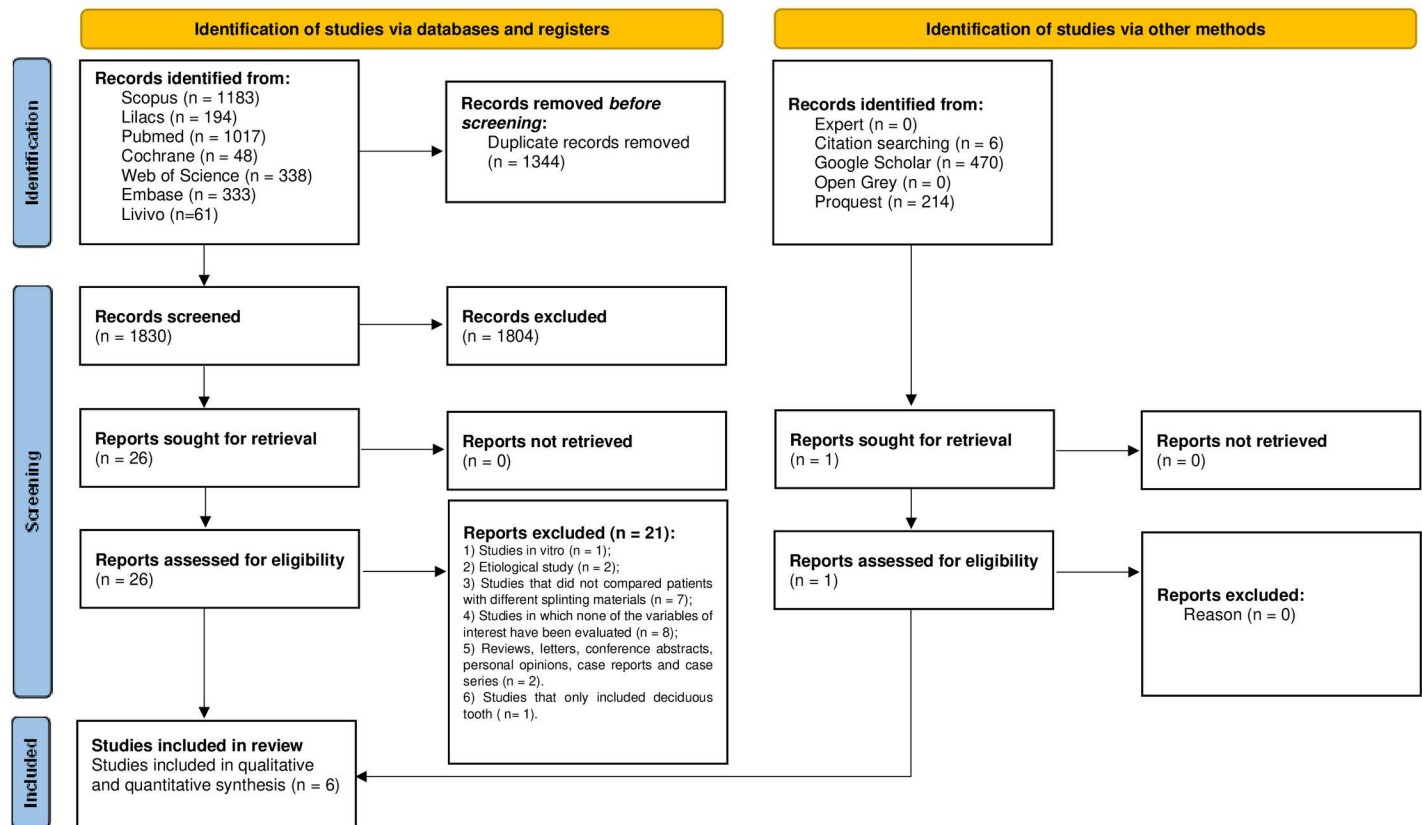

**PRISMA 2020 flow diagram for new systematic reviews which included searches of databases, registers and other sources**

*From:* Page MJ, McKenzie JE, Bossuyt PM, Boutron I, Hoffmann TC, Mulrow CD, et al. The PRISMA 2020 statement: an updated guideline for reporting systematic reviews. BMJ 2021;372:n71. doi: 10.1136/bmj.n71. For more information, visit: http://www.prisma-statement.org/

**Fig 1. Flow chart of literature search and selection criteria.**

fracture location—all of which may affect healing in cases of horizontal intra-alveolar root fracture. The individual results of each study are presented in Table 1.

## Summary of results

There was no difference in healing when comparing the different types of splinting to non-splinting in cases of horizontal intra-alveolar root fracture (Fig 4). The comparisons generated by the network between the different types of splinting are shown in Table 2.

## Exploring inconsistency

The only pair with direct and indirect evidence allowing the inconsistency analysis was the comparison between cap splints, orthodontic bands, and arch bars. No significant difference was found between the direct and indirect evidence (p = 0.977).

## Discussion

The careful monitoring of the degree of mobility and pulp status is essential to ensure a correct and conservative approach in cases of horizontal intra-alveolar root fracture [6]. Therefore, understanding the factors related to the fracture

**Table 1. Characteristics of the included studies.**

| Study, Year (Country) | Sample characteristics | Variables analyzed | Main results | Secondary Results |
|---|---|---|---|---|
| Andreasen et al. 1967 (Denmark) | 50 teeth/48 permanent teeth (32 cap splint/10 archwire/06 no splinting) | Age in years/ Sex/ Type of fracture/ Location of fracture/ Dislocation/ Reduction/ Fixation/ fixation type/ fixation period | No significant difference between the use of splinting | Regarding the time of splinting (immediate/late), it showed that although it did not change the prognosis, it tended to be more expected in teeth that were immediately splinted. The time of splinting also showed no significant difference in the results. Regarding the fracture location, there was no difference in terms of healing. As to fragment displacement, a difference was observed between the types of healing.Fracture reduction was not shown to be a significant variable for healing in these individuals. |
| Zachrisson, Jacobsen 1975 (Norway) | 66 permanent teeth (38 acrylic splint/8 bands and archwire-archbar/7 others) | Localization of fracture/Root development/Dislocation of coronal fragment (Clinical findings)/ Dislocation of coronal fragment (Radiographic findings)/Increased mobility/Concurrent crown fracture/ Tooth contact at closure/Time interval from injury to reduction/Effect of reduction/Time interval from injury to fixation/Fixation type | No significant difference between the use of splinting | The only variable that showed significance regarding the occurrence or not of healing was the degree of displacement of the coronary fragment |
| Andreasen et al. 1989 (Denmark) | 95 permanent teeth (69 acid etch + none/26 orthodontic band) | Luxation of coronal fragment/ Stage of root development/Age of patient at fhe fime of injury/Degree of loosening of coronal fragment/ Separation of fragments (prior to repositioning)/Separation of fragments (after repositioning)/Pulpal sensibility at fhe time of injury/Level of roof fracture/Diameter of the "fracture foramen"/Diameter of the apical foramen/Restorations at the time of injury/Marginal periodontitis/ Type of fixation/Antibiotic therapy/ Injured adjacent teeth | There was no great difference in the occurrence of healing, only in the type. Teeth that were fixed with orthodontic arch have a better chance of repair by connective tissue. | In analyzing the stratified data of this study it was observed that hard tissue repair occurs in teeth with larger apical diameter and less severe dislocation of the coronary fragment, this type of healing did not occur in teeth with compromised periodontal health. In connective tissue healing, only the variables on the presence of restorations and poor periodontal health were significant for the increase in this type of healing. In granulation tissue interposition, the significant variables were: use of orthodontic band, antibiotic use, decreased diameter of the apical foramen and mobility of the coronary fragment. |
| Cvek et al. 2001 (Denmark) | 208 permanent teeth (26 orthodontic bands-archbar/142 cap splints) | Root development/Root development/Diastasis (mm) between/ Fracture location/Fracture type/ Sensibility test/Treatment delay/ Reposition/Fixation/Type of splint/ Fixation length (days) | Healing in hard tissue the stratified and the single variable analysis show that in teeth fixed with the orthodontic band this repair was twice as frequent. | Root formation proved to be an important variable in the healing of fragments by hard tissue, which was observed with greater frequency in teeth with incomplete root formation, and the frequency of this healing decreases with increasing tooth maturity. The displacement of the dental fragment and the degree of separation of the fragments showed that healing in general, is greater when there is no displacement, and likewise, hard tissue healing is greater when there is no displacement. The separation between the fragments before splinting also showed impact on the occurrence or not of healing, but not on the frequency of hard tissue formation. The pulp sensibility right after trauma was relevant in the occurrence or not of healing and in the frequency of hard tissue formation. The repositioning also showed relevance in the occurrence or not of healing and in the formation of hard tissue. The comparison between the two groups did not show differences regarding the type of healing, but there was a higher frequency of hard tissue formation in non-splinted teeth. |

*(Continued)*

**Table 1.** (Continued)

| Study, Year (Country) | Sample characteristics | Variables analyzed | Main results | Secondary Results |
|---|---|---|---|---|
| Cvek et al. 2002 (Denmark) | 94 permanent teeth (56 cap splints/7 comp+metal thread/18 comp+glass ionomer) | Root development/Type of injury/Diastasis (mm) between fragments/Fracture type/Sensibility test/Treatment delay/Reposition/Fixation/ Type of splint/Fixation duration (days) | No significant difference between the use of splinting in the occurrence or type of healing | Teeth with incomplete root formation, and less or no displacement between the coronal fragments, healing with hard tissue formation between the fragments, was more frequent. The pulp sensitivity was an important factor when related to healing, but no relation was found analyzing the pulp sensitivity to the type of healing. Teeth with adequate repositioning presented higher probability of healing with hard tissue formation. |
| Andreasen et al. 2004 (Denmark) | 400 permanent teeth (26 orthodontic band-arch bar/236 cap splint/10 composite/28 composite+arch bar/44 kevlar-glass fiber) | Sex/Age (years)/Root development/ Root development (stage)/ Tooth type/Associated crown fracture/Fracture type/ Fracture location/ Type of injury/ Dislocation/ Diastases (mm)/ Mobility of crown fragment/ Sensibility test/ Examination delay/Reposition/Fixation/ Splint type/Splint mobility/ Fixation start/ Fixation length (days)/ Antibiotics | There is no significant difference between the use of flexible or rigid splinting in the occurrence of healing, whereas in the type of healing, flexible splinting has a better prognosis. | The relevant variables after stratified analysis were: stage of root development when compared to the type of healing and occurrence or not of healing. Fracture location proved to be an important variable in the occurrence or not of healing and fractures located in the cervical region have a lower frequency of necrosis. The displacement of the fragments and the extent of their separation also proved to be significant for the occurrence or not of healing and formation of hard tissue and pulp necrosis. The distance between fragments was important when analyzing the formation of hard tissue and pulp necrosis. The mobility of the fragment was shown to have negative effects on the occurrence of healing, pulp necrosis and hard tissue formation. The pulp sensitivity also showed significance when compared to the occurrence or not of healing, hard tissue formation and frequency of pulp necrosis. The repositioning showed influence in the type of healing, and the administration of antibiotic showed negative effect in the frequency of pulp necrosis and hard tissue formation. |

and the splinting intervention is important to ensure appropriate treatment in such cases. The primary objective of splinting is to prevent any (re)dislocation of the (repositioned) tooth crown without interfering with the healing process of the fractured tooth [11,23]. This study aimed to evaluate which splinting method has a better prognosis in cases of dental trauma involving horizontal intra-alveolar root fractures. Although no significant differences were found among the various splint types, individual study analyses revealed several factors that may influence the prognosis of teeth with horizontal intra-alveolar root fractures. These include gender, age, stage of root development, type of fracture (complex, simple, or partial), fracture location, fragment displacement, repositioning, mobility, duration of splinting, and the use or non-use of systemic medication.

The network meta-analysis revealed no statistically significant difference in relative risk among the different splinting methods, as indicated by credibility intervals that included the null value. According to Cvek et al. (2002) [22], the splint type did not influence the frequency or type of healing. In contrast, Andreasen et al. (2004) [11,23] highlighted the importance of considering fragment displacement, noting that in cases without displacement, splinting did not appear to positively affect fragment fusion or reduce the risk of pulp necrosis The results of this study focused solely on splint type, and it was not possible to evaluate the influence of fragment displacement, splinting duration, age, fracture location, and periodontal health—potential confounding factors previously reported in the literature.

An analysis of secondary outcomes revealed several conditions that may affect the prognosis of teeth with horizontal intra-alveolar root fractures. In all studies, the degree of fragment displacement emerged as a key factor for healing, with

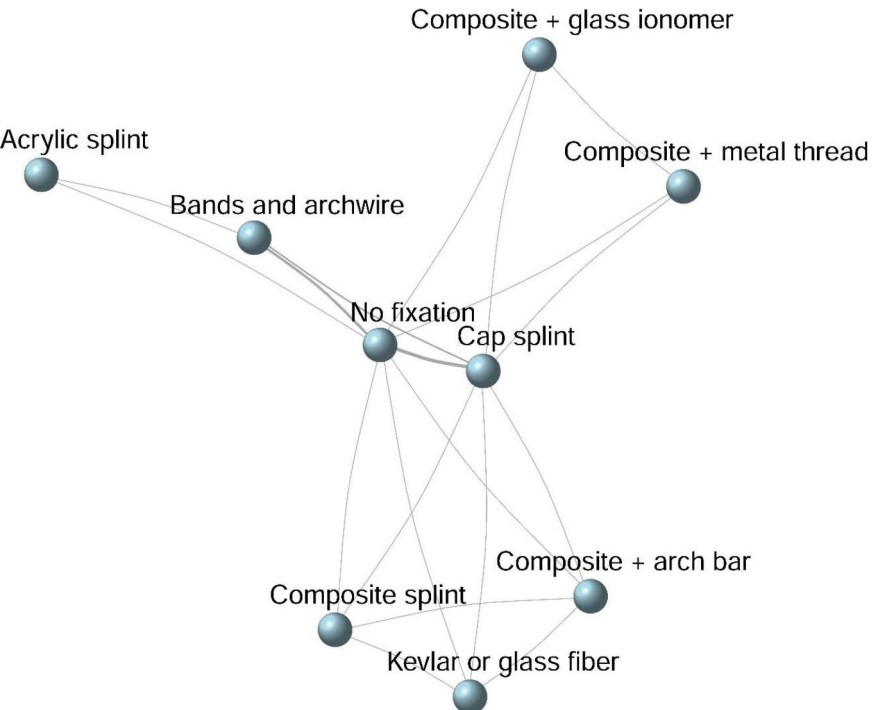

**Fig 2. Network geometry for the healing of cases of horizontal root fracture in different splint types.**

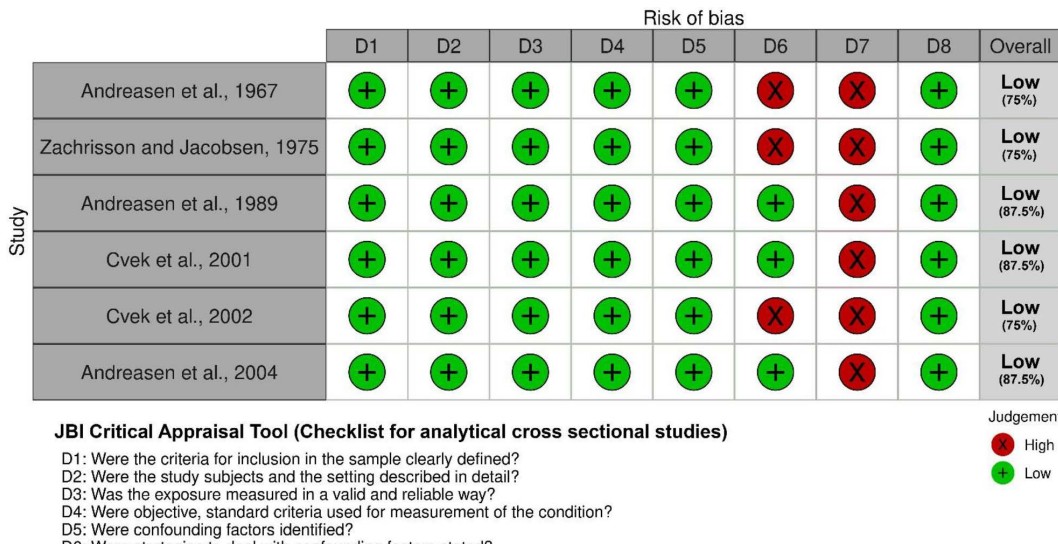

**JBI Critical Appraisal Tool (Checklist for analytical cross sectional studies)**

D1: Were the criteria for inclusion in the sample clearly defined?
D2: Were the study subjects and the setting described in detail?
D3: Was the exposure measured in a valid and reliable way?
D4: Were objective, standard criteria used for measurement of the condition?
D5: Were confounding factors identified?
D6: Were strategies to deal with confounding factors stated?
D7: Were the outcomes measured in a valid and reliable way?
D8: Was appropriate statistical analysis used?

**Fig 3. Assessment of the risk of bias of the studies included in the analysis.** Green indicates a low risk of bias, yellow indicates an uncertain risk of bias, and red indicates a high risk of bias. Summary of risk of bias; Graph.

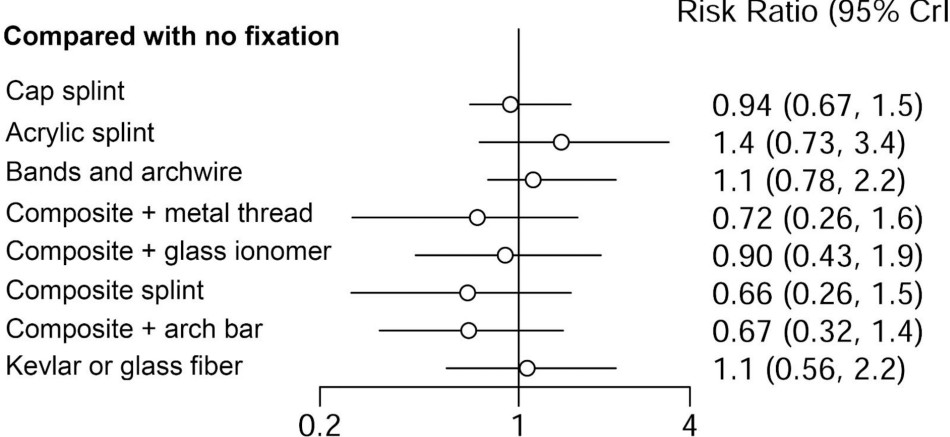

**Fig 4. Forest plot of mixed evidence (direct and indirect) for the relative risk when compared to no fixation for the outcome of healing in cases with horizontal root fracture.**

**Table 2. League table of the evidence generated for the network with the comparison of the relative risk (Crl95%) for scarring between the different splint types.**

| Cap splint | Cap splint | | | | | | | | |
|---|---|---|---|---|---|---|---|---|---|
| **Acrylic splint** | 0.667 (0.284, 1.473) | **Acrylic splint** | | | | | | | |
| **Bands and archwire** | 0.84 (0.463, 1.313) | 1.249 (0.568, 2.627) | **Bands and archwire** | | | | | | |
| **Composite + metal thread** | 1.333 (0.629, 3.683) | 2.035 (0.723, 7.057) | 1.605 (0.734, 4.977) | **Composite + metal thread** | | | | | |
| **Composite + glass ionomer** | 1.044 (0.523, 2.251) | 1.578 (0.602, 4.87) | 1.248 (0.602, 3.261) | 0.785 (0.275, 1.918) | **Composite + glass ionomer** | | | | |
| **Composite splint** | 1.433 (0.68, 3.86) | 2.191 (0.764, 7.667) | 1.727 (0.788, 5.479) | 1.072 (0.307, 3.694) | 1.361 (0.494, 4.453) | **Composite splint** | | | |
| **Composite + arch bar** | 1.417 (0.707, 3.134) | 2.15 (0.787, 6.385) | 1.707 (0.805, 4.388) | 1.066 (0.31, 3.1) | 1.346 (0.495, 3.8) | 0.986 (0.344, 2.438) | **Composite + arch bar** | | |
| **Kevlar or glass fiber** | 0.87 (0.462, 1.91) | 1.308 (0.51, 3.975) | 1.039 (0.539, 2.703) | 0.656 (0.205, 1.919) | 0.836 (0.316, 2.158) | 0.614 (0.216, 1.476) | 0.62 (0.271, 1.441) | **Kevlar or glass fiber** | |
| **No fixation** | 0.936 (0.675, 1.524) | 1.412 (0.728, 3.37) | 1.125 (0.781, 2.197) | 0.716 (0.26, 1.611) | 0.897 (0.434, 1.945) | 0.662 (0.258, 1.526) | 0.667 (0.324, 1.433) | 1.071 (0.558, 2.197) | **No fixation** |

less displacement being associated with a more favorable prognosis. Other important variables included the application of immediate splinting, the degree of fracture reduction, apical foramen diameter, periodontal health, presence of restorations, antibiotic use, fragment mobility, stage of root development, tooth sensitivity after trauma, fracture location, and whether or not splinting was performed. However, not all of these variables were assessed in every study [11,19,21–26].

The position of the fractured fragment in horizontal intra-alveolar root fractures may hinder its visualization in radiographic examinations and may not be adequately detected by two-dimensional imaging techniques [6]. All studies included in this review used two-dimensional methods to assess outcomes. Only one study involved the removal of a fragment from certain teeth for histological analysis [19]. Although radiographic classification of healing types is widely accepted, it may distort results due to the limitations of two-dimensional imaging, which cannot provide the same level of detail as cone beam computed tomography, since radiographs represent three-dimensional structures in only two dimensions.

These factors can compromise the certainty of the generated evidence. Therefore, new studies using three-dimensional imaging combined with statistical analyses capable of controlling for confounding factors are recommended. Moreover, in all studies, statistical analyses were performed for each variable associated with the type of healing and the occurrence or absence of healing. However, only three studies conducted comprehensive analyses that examined the relationships among these variables [11,22,23,26]. The lack of multivariate analysis suggests a methodological limitation that may have distorted the results.

Considering the quality of the studies included in this review, the findings are consistent with the current IADT guidelines, which recommend stabilizing the mobile coronal fragment with passive, flexible immobilization for four weeks. If the fracture is located in the cervical third, a longer immobilization period may be required (up to four months) [14]. The splinting classification performed in this review corroborates the literature [11,23]. The splinting classification used in this review aligns with the existing literature [27]. A second, in vitro study assessed the stiffness of available splints [28]. One of the confounding factors encountered in developing this classification was related to the use of orthodontic wire. These studies indicated that splinting should be considered flexible if the diameter of the orthodontic wire does not exceed 0.3–0.4 mm. Based on this prior research, the classification proposed by Andreasen et al. (2004) was adopted.

Several limitations of this review should be taken into account. These include the fact that the majority of the studies were conducted by the same research group. It is important to highlight that Andreasen was a pioneer in dental traumatology, which may explain the concentration of studies from this group. First, it should be noted that the included studies were primarily cross-sectional observational studies, which limits the ability to draw definitive conclusions. Additionally, the small number of direct comparisons between different splint types further restricts the generalizability of the findings. Another ethical limitation concerns the invasive nature of some outcome assessment methods, such as the histological analyses conducted by Andreasen et al. (1967) [19] Furthermore, is important to note that cone beam computed tomography is not recommended due to its higher radiation dosage compared to periapical radiographs. Consequently, the use of periapical and occlusal radiographs, as recommended by the International Association of Dental Traumatology – IADT [14], remains crucial for accurate diagnosis and assessment of the healing process.

This review presents findings that should be taken into consideration in cases of horizontal intra-alveolar root fracture, including factors associated with changes in prognosis and the type of splint used. Randomized clinical trials would be the most appropriate method for evaluating the effectiveness of different splint types, as they can more effectively control for confounding factors. However, such trials are not feasible for decision-making regarding splint type, as it would be unethical to withhold the appropriate immobilization recommended by the IADT.

As a result, observational cohort studies with appropriate analytical methods to account for variables known to influence healing outcomes and healing type (i.e., potential confounding factors) should be considered. Although meta-regression could offer a more comprehensive analysis, the number of studies included in this review makes such an approach unfeasible. Furthermore, during the study screening and selection process, both Reviewer 1 and Reviewer 2 identified a significant gap in the dental literature—not only regarding horizontal intra-alveolar root fractures but dental trauma in general. It is important to note that the available studies are largely outdated and predominantly originate from a single research group. Nevertheless, the studies included in this review remain highly relevant and continue to serve as key references for the guidelines established by the International Association of Dental Traumatology [19] From the initial phase of study selection, it becomes evident that researching the topic of horizontal intra-alveolar root fractures presents challenges. The epidemiological data mentioned in the introduction clearly illustrates the low incidence of this specific type of trauma [2–5].

## Conclusion

There may be no significant difference in healing outcomes based on the type of splint used for horizontal intra-alveolar root fractures. However, other factors may influence this outcome, such as the degree of fragment displacement, whether

immediate splinting was performed, the degree of fracture reduction, diameter of the apical foramen, periodontal health, presence of restorations, use or non-use of antibiotics, fragment mobility, stage of root development, tooth sensitivity after trauma, and fracture location. Although this study reviewed the existing literature on the topic, the strength of the available evidence remains relatively low, highlighting the need for further research to reinforce the findings in this area.

## Supporting information

**S1 Appendix. Database search strategy.**
(DOCX)

**S2 Appendix. Studies excluded.**
(DOCX)

**S1 File. PRISMA NMA checklist.**
(DOCX)

## Author contributions

**Conceptualization:** Tatiana Carvalho Kowaltschuk, Everdan Carneiro, Odilon Guariza-Filho, Cristiano Miranda de Araujo, Vânia Portela Ditzel Westphalen.

**Data curation:** Tatiana Carvalho Kowaltschuk, Flávio Magno Gonçalves, Bianca Marques de Mattos de Araujo, Natália Teixeira da Silva Cruz, Odilon Guariza-Filho, Cristiano Miranda de Araujo.

**Formal analysis:** Tatiana Carvalho Kowaltschuk, Bianca Marques de Mattos de Araujo, Everdan Carneiro, Cristiano Miranda de Araujo.

**Investigation:** Tatiana Carvalho Kowaltschuk, Natália Teixeira da Silva Cruz, Vânia Portela Ditzel Westphalen.

**Methodology:** Tatiana Carvalho Kowaltschuk, Flávio Magno Gonçalves, Cristiano Miranda de Araujo.

**Project administration:** Tatiana Carvalho Kowaltschuk, Cristiano Miranda de Araujo, Vânia Portela Ditzel Westphalen.

**Resources:** Flávio Magno Gonçalves.

**Software:** Flávio Magno Gonçalves.

**Supervision:** Ulisses Xavier da Silva-Neto, Odilon Guariza-Filho, Cristiano Miranda de Araujo, Vânia Portela Ditzel Westphalen.

**Validation:** Angela Graciela Deliga Schroder.

**Writing – original draft:** Tatiana Carvalho Kowaltschuk, Bianca Marques de Mattos de Araujo, Bruna Belz Antoniazzi, Cristiano Miranda de Araujo.

**Writing – review & editing:** Tatiana Carvalho Kowaltschuk, Bianca Marques de Mattos de Araujo, Angela Graciela Deliga Schroder, Everdan Carneiro, Marisa Nogueira Alencar, Ulisses Xavier da Silva-Neto, Cristiano Miranda de Araujo, Vânia Portela Ditzel Westphalen.

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
