## [Decision Letter · Decision Letter 0]

PONE-D-24-36577SPLINTING IN HORIZONTAL ROOT FRACTURES: A BAYESIAN NETWORK META-ANALYSISPLOS ONE

Dear Dr. Kowaltschuk,

Thank you for submitting your manuscript to PLOS ONE. After careful consideration, we feel that it has merit but does not fully meet PLOS ONE’s publication criteria as it currently stands. Therefore, we invite you to submit a revised version of the manuscript that addresses the points raised during the review process.

We look forward to receiving your revised manuscript.

Kind regards,

Sameh Attia, MS

Academic Editor

PLOS ONE

Journal Requirements:

3. As required by our policy on Data Availability, please ensure your manuscript or supplementary information includes the following:

Reviewers' comments:

Reviewer's Responses to Questions

**Comments to the Author**

1. Is the manuscript technically sound, and do the data support the conclusions?

Reviewer #1: Yes

Reviewer #2: Yes

Reviewer #3: Yes

2. Has the statistical analysis been performed appropriately and rigorously? 

Reviewer #1: Yes

Reviewer #2: Yes

Reviewer #3: I Don't Know

3. Have the authors made all data underlying the findings in their manuscript fully available?

Reviewer #1: Yes

Reviewer #2: Yes

Reviewer #3: No

4. Is the manuscript presented in an intelligible fashion and written in standard English?

Reviewer #1: Yes

Reviewer #2: Yes

Reviewer #3: Yes

5. Review Comments to the Author

Reviewer #1: I have no comments to make, I note that no definitive consideration has been obtained regarding the research question.You can proceed with a re-reading of the language.

In any case, I believe the paper can be accepted in this form.

Reviewer #2: Thanks to the authors for the comprehensive work done. Here are my comments:

1. The abstract needs some corrections in the language and grammar.

2. Please add the registration number of the PROSPERO.

3. Comparison (C) in the inclusion and exclusion. Please make the sentence more like a paragraph.

4. Please add from and till of the years of the articles included in the 'INFORMATION SOURCES AND SEARCH STRATEGY' for example, from 1967 till 2024

5. Table 2. Please adjust the table and add names for the columns and in the footnotes, for example, relative risk (Crl).

6. Please check and add the citation in the submitted draft and correct when needed, especially in the discussion.

Reviewer #3: Dear professor, I have evaluated the meta-analysis study. Articles were not used in the current literature for the meta-analysis study, but I did not come across any current publications on the subject. The specificity of the subject may have caused this result. If the researcher had evaluated studies that evaluated the comparison of splints among themselves in vitro, more recent articles and more studies could have been included in this meta-analysis. The low number of articles also limited the inclusion and exclusion qualifications.

The evidence of the study is therefore inconclusive and the limitations are not adequately explained in the study.

In summary, I am of the opinion that the choice of the topic is not very suitable for meta-analysis.

Good work.

6. PLOS authors have the option to publish the peer review history of their article (what does this mean? ). If published, this will include your full peer review and any attached files.

**Do you want your identity to be public for this peer review?** For information about this choice, including consent withdrawal, please see our Privacy Policy .

Reviewer #1: **Yes: ** Enrico Spinas

Reviewer #2: No

Reviewer #3: No

---

## [Author Response · Author response to Decision Letter 1]

21 Apr 2025

Ref: Submission ID PONE-D-24-36577

Dear Sameh Attia, MS

Academic Editor

PLOS ONE

Thank you very much for your message. We are submitting a revised version of our manuscript after addressing the areas of concern mentioned. The changes to the manuscript are highlighted in red within the document. Please find below our responses to the points raised in your email.

We hope that our corrections are appropriate and that the manuscript may now be reconsidered for publication. Should you have further questions or requests, please do not hesitate to contact us.

Yours sincerely,

Tatiana C Kowaltschuk

Reviewer Comments:

Reviewer #1: I have no comments to make, I note that no definitive consideration has been obtained regarding the research question.You can proceed with a re-reading of the language.

In any case, I believe the paper can be accepted in this form.

Answer: Thank you for your feedback and for noting that the paper can be accepted in its current form. A thorough grammatical review and meticulous revision of the manuscript were performed, and we have ensured our research question is clearly addressed.

Reviewer #2: Thanks to the authors for the comprehensive work done. Here are my comments:

1. The abstract needs some corrections in the language and grammar.

Answer: Thank you for your feedback. We have performed a thorough grammatical review and meticulous revision of the manuscript prior to resubmission.

2. Please add the registration number of the PROSPERO.

Answer: Thank you for your feedback. The PROSPERO registration number has been included in the revised manuscript.

3. Comparison (C) in the inclusion and exclusion. Please make the sentence more like a paragraph.

Answer: We appreciate the comment. The inclusion and exclusion criteria have been revised to improve clarity. The Comparison (C) element is now clearly described in the PICOS paragraph, ensuring coherence in the presentation of the study design.

4. Please add from and till of the years of the articles included in the 'INFORMATION SOURCES AND SEARCH STRATEGY' for example, from 1967 till 2024

Answer: We appreciate the comment. As stated in the Study Design section, no time restrictions were applied during the search. This approach allowed for the inclusion of all available studies on the topic, ensuring a comprehensive and unbiased review of the existing literature.

5. Table 2. Please adjust the table and add names for the columns and in the footnotes, for example, relative risk (Crl).

Answer: Thank you for the suggestion. Table 2 has been revised to include column titles and explanatory footnotes (e.g., relative risk [CrI]), thereby enhancing clarity and aiding interpretation.

6. Please check and add the citation in the submitted draft and correct when needed, especially in the discussion.

Answer: We have carefully reviewed all references and in-text citations, making the necessary corrections to ensure accuracy and proper attribution throughout the manuscript, especially in the discussion section.

Reviewer #3: Dear professor, I have evaluated the meta-analysis study. Articles were not used in the current literature for the meta-analysis study, but I did not come across any current publications on the subject. The specificity of the subject may have caused this result. If the researcher had evaluated studies that evaluated the comparison of splints among themselves in vitro, more recent articles and more studies could have been included in this meta-analysis. The low number of articles also limited the inclusion and exclusion qualifications.

The evidence of the study is therefore inconclusive and the limitations are not adequately explained in the study.

In summary, I am of the opinion that the choice of the topic is not very suitable for meta-analysis.

Good work.

Answer: We appreciate the reviewer’s feedback. In order to ensure the inclusion of the most up-to-date evidence available on the topic, a literature search update was conducted on April 3, 2025, ensuring that the review reflects the most current studies. Regarding the suggestion to include in vitro studies, we chose not to incorporate them, as this is a systematic review with meta-analysis focused on human studies. Including in vitro studies alongside clinical ones would not be methodologically appropriate, especially considering that the outcome assessed refers to the clinical prognosis of lesions treated with different types of splinting. Therefore, this review includes all clinical studies currently available on the subject and highlights the existing gap in the literature — a point that is clearly addressed in the final section of the discussion as a recommendation for future research.

---

## [Decision Letter · Decision Letter 1]

SPLINTING IN HORIZONTAL ROOT FRACTURES: A BAYESIAN NETWORK META-ANALYSIS

PONE-D-24-36577R1

Dear Dr. Kowaltschuk,

We’re pleased to inform you that your manuscript has been judged scientifically suitable for publication and will be formally accepted for publication once it meets all outstanding technical requirements.

Kind regards,

Paolo Boffano

Academic Editor

PLOS ONE

Additional Editor Comments (optional):

Reviewers' comments:

Reviewer's Responses to Questions

**Comments to the Author**

1. If the authors have adequately addressed your comments raised in a previous round of review and you feel that this manuscript is now acceptable for publication, you may indicate that here to bypass the “Comments to the Author” section, enter your conflict of interest statement in the “Confidential to Editor” section, and submit your "Accept" recommendation.

Reviewer #2: All comments have been addressed

Reviewer #4: All comments have been addressed

Reviewer #5: All comments have been addressed

2. Is the manuscript technically sound, and do the data support the conclusions?

Reviewer #2: Yes

Reviewer #4: Yes

Reviewer #5: Yes

3. Has the statistical analysis been performed appropriately and rigorously? 

Reviewer #2: Yes

Reviewer #4: Yes

Reviewer #5: Yes

4. Have the authors made all data underlying the findings in their manuscript fully available?

Reviewer #2: Yes

Reviewer #4: Yes

Reviewer #5: Yes

5. Is the manuscript presented in an intelligible fashion and written in standard English?

Reviewer #2: Yes

Reviewer #4: Yes

Reviewer #5: Yes

6. Review Comments to the Author

Reviewer #2: Dear authors, Thank you for submitting the corrected version of the draft. My comments have been addressed; however, there are a few small comments that need the authors to take care of them.

1. Table 2: Please include footnotes to clarify what the numbers are for. It is included in the table title, but I prefer to mention the relative risk (Crl 95%) in the table or in the footnote.

2. Figure 3: The authors mentioned A and B (graph). Please check again.

3. Remove the other information section after the conclusion, as it is mentioned before.

Reviewer #4: We gladly accept articles that are original, well-written, and relevant. Submissions must follow our guidelines and editorial standards.

Reviewer #5: (No Response)

7. PLOS authors have the option to publish the peer review history of their article (what does this mean? ). If published, this will include your full peer review and any attached files.

**Do you want your identity to be public for this peer review?** For information about this choice, including consent withdrawal, please see our Privacy Policy .

Reviewer #2: No

Reviewer #4: **Yes: ** andrea melle

Reviewer #5: **Yes: ** Francesca Santeusanio

---

## [Editor Report · Acceptance letter]

PONE-D-24-36577R1

PLOS ONE

Dear Dr. Kowaltschuk,

I'm pleased to inform you that your manuscript has been deemed suitable for publication in PLOS ONE. Congratulations! Your manuscript is now being handed over to our production team.

Kind regards,

on behalf of

Dr. Paolo Boffano

Academic Editor

PLOS ONE